# Development of “Mathematical Technology for Cytopathology,” an Image Analysis Algorithm for Pancreatic Cancer

**DOI:** 10.3390/diagnostics12051149

**Published:** 2022-05-05

**Authors:** Reiko Yamada, Kazuaki Nakane, Noriyuki Kadoya, Chise Matsuda, Hiroshi Imai, Junya Tsuboi, Yasuhiko Hamada, Kyosuke Tanaka, Isao Tawara, Hayato Nakagawa

**Affiliations:** 1Department of Gastroenterology and Hepatology, School of Medicine, Mie University, Tsu 514-8507, Japan; y-hamada@clin.medic.mie-u.ac.jp (Y.H.); nakagawah@med.mie-u.ac.jp (H.N.); 2Department of Molecular Pathology, Osaka University, Osaka 565-0871, Japan; k-nakane@sahs.med.osaka-u.ac.jp; 3Department of Radiation Oncology, School of Medicine, Tohoku University, Sendai 980-8577, Japan; kadoya.n@rad.med.tohoku.ac.jp; 4Department of Pathology, Mie University Hospital, Tsu 514-8507, Japan; sechico14@med.mie-u.ac.jp (C.M.); qchan@med.mie-u.ac.jp (H.I.); 5Department of Endoscopic Medicine, Mie University Hospital, Tsu 514-8507, Japan; t-junya0128@clin.medic.mie-u.ac.jp (J.T.); kyosuket@med.mie-u.ac.jp (K.T.); 6Department of Hematology and Oncology, School of Medicine, Mie University, Tsu 514-8507, Japan; itawara@clin.medic.mie-u.ac.jp

**Keywords:** artificial intelligence, benign tissue, diffusion coefficient, Mathematical Technology for Cytopathology, medical image, multivariate analysis, nuclear boundary, pancreatic ductal adenocarcinoma, rapid on-site evaluation, reaction–diffusion system

## Abstract

Pancreatic ductal adenocarcinoma (PDAC) is a leading cause of cancer-related death worldwide. The accuracy of a PDAC diagnosis based on endoscopic ultrasonography-guided fine-needle aspiration cytology can be strengthened by performing a rapid on-site evaluation (ROSE). However, ROSE can only be performed in a limited number of facilities, due to a relative lack of available resources or cytologists with sufficient training. Therefore, we developed the Mathematical Technology for Cytopathology (MTC) algorithm, which does not require teaching data or large-scale computing. We applied the MTC algorithm to support the cytological diagnosis of pancreatic cancer tissues, by converting medical images into structured data, which rendered them suitable for artificial intelligence (AI) analysis. Using this approach, we successfully clarified ambiguous cell boundaries by solving a reaction–diffusion system and quantitating the cell nucleus status. A diffusion coefficient (*D*) of 150 showed the highest accuracy (i.e., 74%), based on a univariate analysis. A multivariate analysis was performed using 120 combinations of evaluation indices, and the highest accuracies for each *D* value studied (50, 100, and 150) were all ≥70%. Thus, our findings indicate that MTC can help distinguish between adenocarcinoma and benign pancreatic tissues, and imply its potential for facilitating rapid progress in clinical diagnostic applications.

## 1. Introduction

Pancreatic ductal adenocarcinoma (PDAC) is a leading cause of cancer-related death worldwide; the 5-year survival rate for patients with PDAC is less than 10%, and most patients die within 2 years after diagnosis [1,2]. Although surgery is recommended for patients with early-stage or locally advanced disease, less than 20% of such patients are good candidates for resection. Because PDAC is a systemic disease, multimodal treatment is required, such as neoadjuvant/adjuvant chemotherapy and chemo radiation therapy [3]. Therefore, obtaining a definitive diagnosis by endoscopic ultrasonography-guided fine-needle aspiration cytology (EUS-FNA) and endoscopic ultrasonography-guided fine-needle biopsy (EUS-FNB) before surgery or treatment has become increasingly essential [4,5].

When making a definitive diagnosis of PDAC, a cytological diagnosis by EUS-FNA can be strengthened by performing a rapid on-site evaluation (ROSE), which helps to provide immediate feedback and enables a diagnosis to be made in the shortest possible time [6,7,8]. However, some issues exist with ROSE, such as the limited number of facilities where ROSE can be provided [9]. One reason for this may be the relative lack of cytologists who can immediately diagnose pancreatic cancer with ROSE; thus, the burden on cytologists is increasing. Therefore, there is a need to develop a new diagnostic-support technology.

Currently, attempts are being made to build automated systems using artificial intelligence (AI) [10,11]. However, AI itself has problems, such as high cost, low versatility (since the results depend on the quality and quantity of the teaching data), and an unknown reason for the diagnostic result (AI’s black box). In addition, the distribution of the cell nucleus (which provides a clue for diagnosis) is random. Thus, the random distribution of the cell nucleus makes it difficult to apply algorithms based on the supervised data used in AI. Therefore, using AI for cytological diagnosis is difficult, because the data need to be analyzed by capturing the random and three-dimensional distribution of lesions.

In contrast, the mathematical method developed in this study does not require teaching data or a large-scale computer system. Our mathematical method clarifies ambiguous cell boundaries by solving certain differential equations. Specifically, we (I) clarify ambiguous nuclear boundaries by solving a reaction–diffusion system and (II) quantitatively evaluate the cell nucleus status using mathematical principles, an approach known as the homology profile (HP) method, to match it with physicians’ interpretations [12,13,14,15]. The HP method is an algebraic tool for measuring the topological features of objects [16]. Given a topological space, the HP algorithm computes the number of connected components and holes using the structure of that space, based on continuous thresholds. Recently, Qaiser et al. employed the HP algorithm for tumor segmentation by focusing on the connectivity between nuclei [12,17].

Using the HP method, medical images are converted into structured data, which renders them effective for use with AI techniques. We named the series of methods developed in this study as the Mathematical Technology for Cytopathology (MTC) algorithm. The essence of MTC is the structuring of medical images; once the image data are structured, the bottleneck of applying AI technology to pathological images (a current limitation) will be largely eliminated, leading to rapid progress in clinical applications. MTC does not require a large-scale computational system and does not depend on monitoring data, such as staining conditions. MTC can be applied universally because it is robust. In addition, the algorithm is clear and the reason for the diagnosis can be explained. In this study, we investigated the applicability of MTC to support the cytological diagnosis of pancreatic cancer.

## 2. Materials and Methods

### 2.1. Study Design

This study was designed as an exploratory observational study to analyze whether PDAC can be diagnosed using MTC with medical records and existing cytology specimens, without involving invasion or intervention. This research was conducted as a joint effort between investigators at Mie University, Osaka University, and Tohoku University. We attempted to differentiate adenocarcinoma from benign pancreatic cells by quantitatively analyzing information on the distributions (size and variability of the cell cluster form) of cells and cell nuclei via MTC analysis of cytological images (103 normal and 143 adenocarcinoma specimens) obtained by EUS-FNA or EUS-FNB.

### 2.2. Procedure of EUS-FNA/EUS-FNB and Diagnosis

A convex-array echo-endoscope (GF-UCT260, Olympus, Tokyo, Japan) was used for EUS-FNA and EUS-FNB procedures. After identifying tumors using B-mode imaging and confirming the absence of vessels in the target area, we punctured the pancreatic mass under endoscopic ultrasonographic guidance. We mainly used four types of needles, namely, 25 G and 22G FNA needles (EZ-shot 3 Plus, Olympus, Tokyo, Japan), and 22G and 19G Franseen needles (Acquire, Boston Scientific, Natick, MA, USA). The different needles were used according to their availability.

A cytologist immediately examined each specimen with ROSE using rapid staining (Diff-Quick stain; International Re-agents, Kobe, Japan) to verify that a sufficient sample was obtained. Further punctures were performed in cases where an insufficient sample was obtained. We confirmed the diagnosis of PDAC by cytological and/or histological analyses with EUS-FNA and EUS-FNB specimens. Both the cytological and pathological diagnoses were based on the review of all these materials by cytopathologists.

### 2.3. Automatic Diagnosis Assistance System

To automate the ROSE analysis of each series of contents, it was necessary to analyze the morphology and arrangement information of the “nucleus,” such as the irregularity of the cell nucleus. It was only necessary to extract the nuclei; however, the nuclei were layered on top of each other, which made it difficult to separate them using ordinary image processing methods. Therefore, only the cell nuclei were extracted using the “reaction–diffusion method,” which involves separation by adjusting the gray areas to either black or white (Figure 1). At present, the “reaction–diffusion method” takes approximately one minute per image. With further improvements, it should be possible to process each image in approximately 10 s.

### 2.4. The Mathematical Method

Reaction–diffusion systems are often used to analyze self-organization phenomena [18], but they can also be applied to image analysis. Here, the method was applied to detect ambiguous boundaries of the nuclei. In general, physicians detect nuclei by ignoring small particles and light-colored areas. However, this method shows poor reproducibility when performing ordinary image analysis. We sought to increase the reproducibility by solving a reaction–diffusion system. The most important key is Equation (3), which is presented below in Section 2.1. The first term (the reaction term) changes brightly colored areas to black or white. The second term (the diffusion term) serves as an averaging factor, causing small particles to disappear. The associated mechanism can be explained as follows: if the value of u is in interval (i), then the reaction term is negative (Equation (3)). If $u_{t}$ is considered negative, then the value of u decreases. Conversely, if u is in interval (ii), then the value of u increases (Figure 2). Here, let interval (i) be (a, 0) and interval (ii) be (0, b). Therefore, the value of u finally converges to that of *a* (black) or *b* (white). Figure 1 shows representative reaction–diffusion results (right panels), which seem to be close to the cytological images (left panels).

The idea of applying a reaction–diffusion system to image analysis was first introduced and developed by Nomura et al. for detecting edges in images with variable brightness [19]. In addition, Mahara et al. applied this method to detect vague boundaries, such as material grains (JIS-SUJ2) and capillaries at the base of the fingernails (Figure 3) [20].

#### 2.4.1. The Reaction–Diffusion System

Local average thresholds were determined based on the following diffusion equation:(1)$$\frac{\partial a}{\partial t}=D_{a}\nabla^{2}a$$
where *D_a_* is the diffusion coefficient. The value *a* is a threshold for running the FitzHugh–Nagumo (FHN) equations, i.e., Equations (3) and (4) [19,21]. The initial value *a*_0_ of *a* is determined by the following:(2)$$a_{0}=0.15+0.2\;\frac{\left(I-I_{\min}\right)}{\left(I_{\max}-I_{\min}\right)}\;$$
where *I* is the brightness of pixels with the gray scale (0–255) in the original image. *I*_max_ and *I*_min_ are the maximum and minimum brightness values of the pixels, respectively.

Next, a reaction–diffusion system was used based on the FHN equations. The system is described by Equations (3) and (4):(3)$$\frac{\partial u}{\partial t}=\frac{1}{\epsilon }\left[u\left(1-u\right)\left(u-a\right)-v\right]+D_{u}\nabla^{2}u$$
(4)$$\frac{\partial v}{\partial t}=u-bv+D_{v}\nabla^{2}v$$
where *D_u_* and *D_v_* are the diffusion coefficients for the variables *u* and *v*, respectively. The parameter ε is a positive small constant (0 < ε < 1). The parameter *b* is a positive constant and is spatially homogeneous. The initial value of *u* is defined as follows:(5)$$u_{0}=0.15+\left(C-0.15\right)\;\;\frac{\left(I-I_{\min}\right)}{\left(I-I_{\max}\right)}$$
where *C* is constant, and the appropriate value of this parameter depends on the original image used for edge detection. The initial value of *v* was set to zero uniformly in this domain.

#### 2.4.2. Numerical Computations

Numerical calculations were carried out as previously described [18,19,20,22,23]. The initial conditions of *a* and *u* were determined based on the pixel data of the images (Equations (2) and (5)). We discretized Equations (1), (3) and (4), and used the fourth-order Runge–Kutta method for space and the finite-difference method for time. The Neumann boundary conditions were applied. 

Initially, Equation (1) was calculated using *t* = 0.005 and diffusion coefficient (*D_a_*) = 2500.0. Next, Equations (1), (3) and (4) were calculated with t in the interval [0.005, 0.02]. The parameter values were ε = 0.0002 and *b* = 20.0. Our numerical computation stopped after approximately 4–5 s, using an ordinary laptop computer. The final result of *u* was translated to the binary images.

### 2.5. Calculating the Quantitative Index

The quantitative index was calculated for the connected components with areas of 100–1000 pixels in the images. The connected components with areas outside of this range were considered to be noise or to reflect instances where the nuclear boundary could not be distinguished well. Seven quantitative indexes were calculated, including the following: (1) number of pixels; (2) area (pixels); (3) interquartile range of the area; (4) area/pixel; (5) average perimeter of the connected components; (6) average circularity of the connected components; and (7) interquartile circularity range of the connected components. The parameter *D* (induced by *D_u_* and *D_v_*) can be regarded as reflecting the state of the tissue staining. In this study, we selected three parameters for *D* (50, 100, and 150). Important indices for detecting adenocarcinoma cells were identified by calculating the accuracy, sensitivity, and specificity of the quantitative index.

### 2.6. Classifying Tissues as Normal or Adenocarcinoma Tissues

Tissue classifications (i.e., normal or adenocarcinoma) were performed using univariate and multivariate analysis with the seven quantitative indexes mentioned in Section 2.5.

When performing univariate analysis, the median value was used as the threshold value to perform the classification. For multivariate analysis, we combined the quantitative indexes by summing them. It should be noted that we repeatedly performed multivariate analysis by changing the number of combined quantitative indexes from two to seven (i.e., 120 combinations). In this analysis, min–max normalization was used to normalize each quantitative index. MATLAB R2020a (Math Works, Natick, MA, USA) was used for the calculation.

### 2.7. Evaluating the Classification Accuracy

The accuracy, sensitivity, and specificity were calculated by the following equations:(6)$$\mathrm{Accuracy}\;=\;\frac{TP+TN}{TP+TN+FP+FN}$$
(7)$$\mathrm{Sensitivity}\;=\;\frac{TP}{TP+FN}$$
(8)$$\mathrm{Specificity}\;=\;\frac{TN}{TN+FP}$$
Here, *TP*, *TN*, *FP*, and *FN* represent the true-positive, true-negative, false-positive, and false-negative values, respectively. MATLAB R2020a was used for evaluating the accuracy of the classification method.

## 3. Results

Here, *D* was calculated as an index that depended on the diffusion coefficients D*u* and D*v*. First, to select the appropriate parameters, we outputted the reaction–diffusion images with three different *D* values (50, 100, and 150). As the *D* value decreased, the unnecessary parts of the edges became visible, instead of the core content. In contrast, as the *D* value increased, the unnecessary parts of the edges disappeared, while the content showed a tendency to almost disappear. Therefore, we selected three *D* values with acceptable performance: *D* = 50, *D* = 100, and *D* = 150 (Figure 4).

For all the images, the quantitative indexes were extracted by cropping around the selected cell masses. With the univariate analysis, the highest accuracies for each parameter were 71% (*D* = 50, number of pixels; Table 1), 69% (*D* = 100, interquartile range of circularity of the connected components; Table 2), and 74% (*D* = 150, interquartile range of circularity of the connected components; Table 3), respectively. These results showed that setting *D* to 150 resulted in the highest accuracy (i.e., 74%) among all three parameters studied.

The multivariate analysis was performed using 120 combinations of evaluation indices. With the multivariate analysis, the highest accuracies for each parameter were 75% (*D* = 50, number of pixels + interquartile area range + average perimeter of the connected components; Table 1), 70% (*D* = 100, number of pixels + interquartile area range; Table 2), and 74% (*D* = 150, area/pixel + interquartile circularity range of the connected components; Table 3), respectively.

## 4. Discussion

The results of this study show that MTC could be used to distinguish between adenocarcinoma tissue and benign pancreatic tissue. Although MTC showed excellent results in discriminating adenocarcinoma from benign patterns in cytology images, there were three problems. One problem was that the edges of the cell clusters often overlapped with each other, making it difficult to capture individual nuclei, resulting in false-positive results. To solve this problem, the cytologist manually selected the region of interest to remove the unnecessary overlap. The second problem was related to the nuclear area in aggregated pancreatic cancer cells with mucus production. Although the distance between the nuclei was irregular, due to the wider cytoplasm of mucus-producing cancer cells, nuclear enlargement appeared to be relatively mild and contributed to the false-negative results. The last problem was that little information was available when the specimens were small and, thus, the judgments varied. The latter two problems can be solved by increasing the number of patterns and performing deep learning.

To the best of our knowledge, no studies have used machine learning or deep learning to support the cytological analysis of pancreatic tissues to diagnose adenocarcinoma in pancreatic EUS-FNA specimens. In 2021, Naito et al. reported the first application of deep learning to detect adenocarcinoma in pancreatic EUS-FNB specimens [24]. They stated that the specimens that pathologists needed for diagnosing adenocarcinoma included various tissue components, such as invasive ductal carcinoma cells in desmoplastic stromata and circulating fragmented and intact cancer cells in the blood. Histological diagnosis is based on the diagnosis of both cellular and structural atypia, whereas cytological diagnosis is based on the morphological abnormalities of individual cells, such as nuclear atypia. Therefore, it is more difficult to directly incorporate cytological diagnosis into deep learning than it is to incorporate histological diagnosis.

Recently, EUS-FNB has been used more than EUS-FNA for tissue acquisition [25,26,27], as EUS-FNB has been reported to provide more stable diagnostic results after improvements were made to the puncture needle [28,29]. However, while EUS-FNB is useful for diagnosing large masses, it is quite difficult to collect tissue fragments by EUS-FNB from small masses (i.e., <1 cm in diameter). In such cases, cytology by EUS-FNA with ROSE may often be more useful. Mie et al. reported that EUS-guided tissue acquisition from small solid pancreatic lesions for ROSE had a high diagnostic yield and was safe [30]. Similarly, in our institution, when tissue samples are obtained under EUS from a small pancreatic mass, they are confirmed by ROSE, and the samples are processed as direct smears and/or formalin-fixed core biopsy specimens.

The role of cytologists in ROSE is significant. Fitzpatrick et al. reported the diagnostic performance of cytopathology (CP) in the evaluation of pancreatic EUS-FNB specimens, evaluated the FNB diagnostic performance stratified by tissue triage, and reviewed the specimen types [31]. They reported that CP accurately diagnosed pancreatic FNB specimens, the ROSE review by CP improved the diagnostic yield and operating characteristics, and that a concurrent review of both the cytological features of direct smears and the architectural features of core biopsies improved the overall diagnostic performance. These findings highlighted the importance of CP in assessing FNB specimens to evaluate the adequacy and render a preliminary diagnosis at the time of the procedure. Therefore, it is important to train cytologists to perform ROSE quickly and accurately. The development of this diagnostic aid technology, MTC, is expected to be very useful in clinical practice, and serve as a good teaching tool for training cytologists, by comparing the analytical results of MTC with their own diagnoses.

Our study has several limitations. One limitation is that the test images were all obtained from a single institution; therefore, it is uncertain how well the MTC model would perform with images obtained from a different institution. Second, no external validation was performed. The third limitation is that the test set size was small (103 normal specimens and 143 adenocarcinoma specimens), and it might not include all the potential variations in cases that could be encountered. As future work, we intend to further develop and evaluate our model with multiple test sets obtained from different medical institutions, to assess its generalization performance and move closer towards adopting such assistive models in routine cytological diagnosis workflows.

## 5. Conclusions

In the future, we can expect to improve the accuracy by selecting optimal parameters for each extracted image. Because MTC is simple and does not require supervisory data, it can be applied to various medical facilities and is expected to be useful for diagnosis support in the future.

## Figures and Tables

**Figure 1 diagnostics-12-01149-f001:**
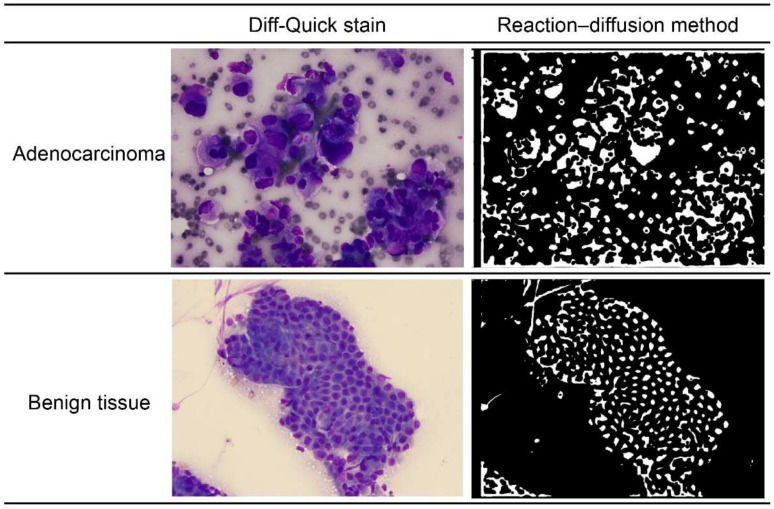
The left panels: Diff-Quick stain of adenocarcinoma and benign pancreatic tissue. The right panels: “reaction–diffusion method” of adenocarcinoma and benign pancreatic tissue.

**Figure 2 diagnostics-12-01149-f002:**
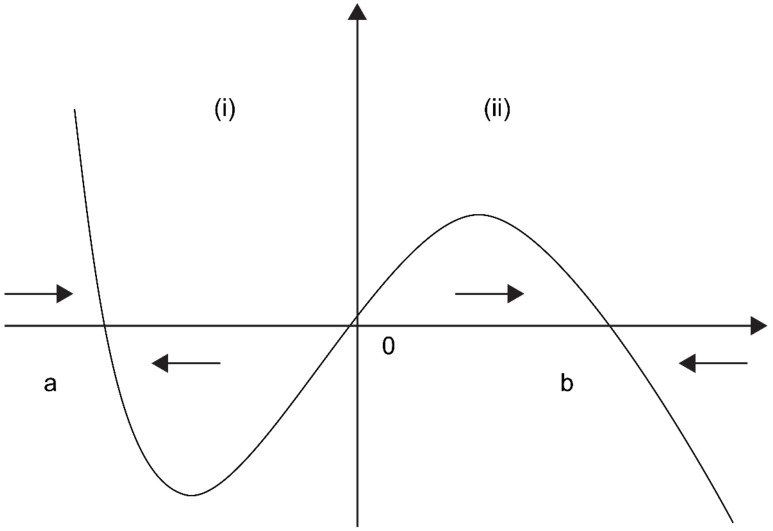
If the value of u is in interval (**i**), then the reaction term is negative. If $u_{t}$ is considered negative, then the value of u decreases. Conversely, if u is in interval (**ii**), then the value of u increases. Here, let interval (**i**) be (a, 0) and interval (**ii**) be (0, b). Therefore, the value of u finally converges to that of *a* (black) or *b* (white).

**Figure 3 diagnostics-12-01149-f003:**
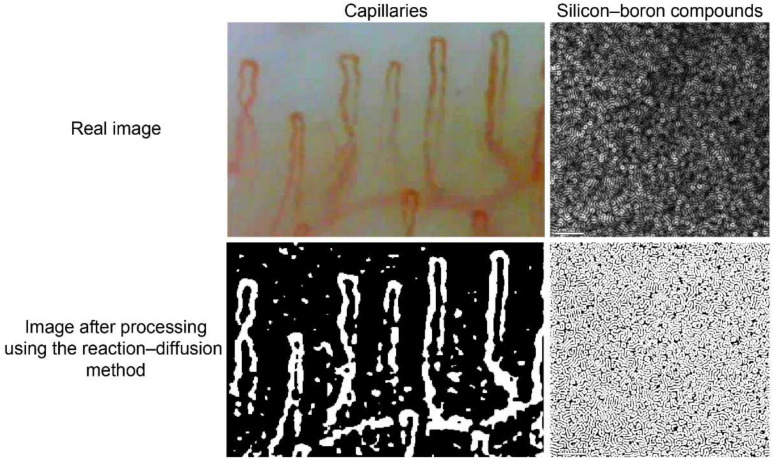
The left panels show the real image of capillaries at the base of the fingernails and the image after processing using the reaction–diffusion method. The right panels show the real image of silicon–boron compounds and the image after processing using the reaction–diffusion method.

**Figure 4 diagnostics-12-01149-f004:**
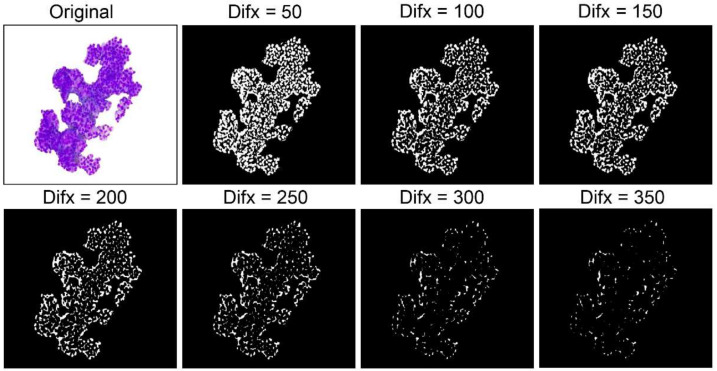
The reaction–diffusion images with different *D* values: As the *D* value decreased, the unnecessary parts of the edges became visible, instead of the core content. In contrast, as the *D* value increased, the unnecessary parts of the edges disappeared, while the content showed a tendency to almost disappear.

**Table 1 diagnostics-12-01149-t001:** Univariate and multivariate analysis (D-50).

	Quantitative Index	Accuracy (%)	Sensitivity (%)	Specificity (%)
Univariate analysis	Number of pixels	71	75	65
	Area	39	47	28
	Interquartile area range	67	71	61
	Area/pixel	68	72	63
	Average perimeter of the connected components	57	64	48
	Average circularity of the connected components	46	53	36
	Interquartile circularity range of the connected components	43	50	33
Multivariate analysis	Number of pixels + interquartile area range + average perimeter of the connected components	75	78	70

**Table 2 diagnostics-12-01149-t002:** Univariate and multivariate analysis (D-100).

	Quantitative Index	Accuracy (%)	Sensitivity (%)	Specificity (%)
Univariate analysis	Number of pixels	68	72	62
	Area	42	49	31
	Interquartile area range	67	71	61
	Area/pixel	66	71	60
	Average perimeter of the connected components	40	49	27
	Average circularity of the connected components	24	34	10
	Interquartile circularity range of the connected components	69	74	64
Multivariate analysis	Number of pixels + interquartile area range	70	74	65

**Table 3 diagnostics-12-01149-t003:** Univariate and multivariate analysis (D-150).

	Quantitative Index	Accuracy (%)	Sensitivity (%)	Specificity (%)
Univariate analysis	Number of pixels	56	62	48
	Area	52	58	44
	Interquartile area range	69	73	64
	Area/pixel	55	61	48
	Average perimeter of the connected components	46	54	34
	Average circularity of the connected components	23	33	9
	Interquartile circularity range of the connected components	74	78	69
Multivariate analysis	Area/pixel + interquartile circularity range of the connected components	74	77	70

## Data Availability

Not applicable.

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
