# Peer review of "Development of “Mathematical Technology for Cytopathology,” an Image Analysis Algorithm for Pancreatic Cancer"

_diagnostics, 2022, doi:10.3390/diagnostics12051149_

Round 1
Reviewer 1 Report
I am satisfied with the clinical and cytopathological aspects of this article but it needs input from a biostatistics expert to check the validity of the mathematical model.
Author Response
Thank you for your meaningful comment.
Dr. Nakane K. is the expert who established this mathematical model.
We recognize the validity of the mathematical model since this formula is a well-established method as shown in the cited paper.
Reviewer 2 Report
The manuscript entitled "Development of "Mathematical Technology for Cytopathology," an Image-Analysis Algorithm for Pancreatic Cancer" highlighted that MTC can help distinguish between adenocarcinoma and benign pancreatic tissues and imply its potential for facilitating rapid progress in clinical diagnostics applications.
- The Authors should provide the expand forms for all acronyms through the text when they first appear.
Author Response
Thank you for your meaningful comment.
We checked that we provided the expand forms for all acronyms through the text when they first appear.
Reviewer 3 Report
General remarks
In the present paper, the authors applied the Mathematical Technology for Cytopathology (MTC) algorithm to diagnose pancreatic ductal adenocarcinoma cases. They noted that MTC can help distinguish between adenocarcinoma and benign pancreatic tissues. The study has a major role in USG guided fine needle aspiration cytology of pancreatic ductal carcinoma.
Specific remarks
- The authors mentioned how to calculate the diffusion coefficient of the cell clusters. The method to assess the chromatin pattern should be mentioned.
- Truly speaking the authors mentioned chromatin status on page 1, line 30. However, nothing is mentioned in the status of the chromatin in the main text.
- The authors mentioned that the study will help in the rapid on-site assessment of cell clusters. Is it logical? How much time is taken in each case to assess the diffusion coefficient of the cell clusters? Please mention the time.
Author Response
Thank you for your meaningful comment.
Our method assess the cell nucleus pattern. Thus, we changed the sentences.
1. Using this approach, we successfully clarified ambiguous cell boundaries by solving a reaction–diffusion system and quantitating the cell nucleus status (line 29).
2. In addition, the distribution of the cell nucleus (which provides a clue for diagnosis) is random. Thus, the random distribution of the cell nucleus makes it difficult to apply algorithms based on supervised data used in AI (line 59-62).
3. Specifically, we (I) clarify ambiguous nuclear boundaries by solving a reaction–diffusion system and (II) quantitatively evaluate the cell nucleus status using mathematical principles, an approach known as the homology profile (HP) method, to match it with physicians’ interpretations (line 66-69).
We also mentioned the time of the reaction-diffusion method. At present, the "reaction-diffusion method” takes approximately one minute per image. With further improvements, it should be possible to process each image in approximately 10 seconds (line 119-121).Reviewer 4 Report
Dear authors,
Even if I am not qualified to review the mathematical and computational methods, I have read with interest your paper and I like it. My only suggestion is to add a distinct ”5. Conclusion” section, because the interesting conclusion of your work needs a proper emphasizing chapter.
Good luck!
Author Response
Thank you for your kind comments. I appreciate a lot.
I added the conclusion.
5. Conclusion
In the future, we can expect to improve the accuracy by selecting optimal parameters for each extracted image. Because MTC is simple and does not require supervisory data, it can be applied to various medical facilities and is expected to be useful for diagnosis support in the future (line 290-295).